# Multiple Sclerosis: An Ethnically Diverse Disease with Worldwide Equity Challenges Accessing Care

**DOI:** 10.3390/neurolint18010002

**Published:** 2025-12-22

**Authors:** Victor M. Rivera

**Affiliations:** Neurology Department, Baylor College of Medicine, Houston, TX 77030, USA; vrivera@bcm.edu

**Keywords:** multiple sclerosis, genetics, healthcare disparities, social determinants of health, prevalence, disease modifying therapies, access to therapy

## Abstract

Multiple sclerosis (MS) affects approximately 2.9 million people in the world, exerting a significant economic and societal burden. The disease is increasingly identified among populations considered as uncommonly affected. MS is reported in all regions of the World Health Organization (WHO) member states in Africa, the Americas, South-East Asia, Europe, the Eastern Mediterranean and the Western Pacific, affecting all ethnicities while exhibiting substantially variable prevalences. Countries with high MS prevalence and some with moderate frequencies generally have economically better structured healthcare systems. Nevertheless, health disparities in these countries are accentuated by suboptimal accessibility of care for their minorities, immigrants and other underserved populations. Social Determinants of Health (SDOH) might have an impact on morbidity and higher rates of disability. Large segments of the world population (i.e., African, Latin American, people from the Middle East and Southeast Asia) do not have access to adequate MS diagnostic procedures, compounded by reduced availability of neurologists. Healthcare disparities exist practically in every country of the world. Active wars and a large number of refugees resulting from conflict augments the challenges to healthcare systems. These global factors constitute obstacles to the adequate management of MS. A collective international path is required to facilitate access to highly effective, albeit onerous treatments, some already approved and being utilized, i.e., monoclonal antibodies and B-lymphocyte depletory agents, and others foreseen in the future as advanced therapeutic molecules continue to develop.

## 1. Introduction

Multiple Sclerosis (MS), a CNS multifactorial disease, develops in a genetically susceptible individual exposed to multiple environmental influences while immune mechanisms mediate inflammatory, demyelinating and neurodegenerative processes. Among the reported environmental factors, geographic latitude, insufficient solar radiation, low vitamin D levels, child obesity and smoking have been cited [1]. The disease is the most common cause of non-traumatic neurological disability in young adults (specified as under age 45) [2]; it is more frequent in women with a ratio of 2–3/1, or higher in certain areas, and has a worldwide distribution with variable frequencies [3]. The MS International Federation (MSIF) reports 2.9 million people living with MS in practically all areas of the world. Higher prevalence is recognized in distinct areas inhabited by white (“Caucasian”) populations: Europe, North America, Australia and New Zealand. Recent increases have been noted in some countries in the Middle East and Latin America. According to the MSIF, the disease prevalence is low in Southeast Asia, Africa and the Western Pacific, but there is no data reported from some Central Asian, Central and West African countries, Greenland, Madagascar, Jordan, Vietnam, Philippines and Papua New Guinea [4]. The strongest genetic risk factor occurring in almost two thirds of people with MS is constituted by the HLA-DRB1*15:01, although this inheritable marker varies significantly by population and ethnicity [5]. MS is a chronic, onerous disease. Its economic gravamen initiates with the complexities of the diagnostic process and later the cumulative burden of high-priced disease modifying therapies (DMT) and the complexity of management in general. These costs are compounded by direct and indirect expenses, comorbidities, and disability. Healthcare systems from low and medium-income countries are also usually ranked as areas with low MS prevalence, confronting significant economic barriers and logistic national challenges addressing these concerns [6]. An actual example of increasing costs posed by the disease process involves the elements required for initial diagnosis, adhering to the last version of the McDonald criteria 2024 including magnetic resonance imaging (MRI), cerebrospinal fluid (CSF) analysis and tests for optic nerve pathology such as visual evoked responses (VER) and optic coherent tomography (OCT) studies [7].

Countries with high MS prevalence and some with moderate frequencies are, in general, economically better structured in their public health systems, and people may have access to private insurance coverage. Negotiations with the pharmaceutical industry and diverse healthcare stakeholders are possible. Nevertheless, health disparities still notably exist in these countries, accentuated by suboptimal access to care affecting ethnic and racial minorities, immigrants and other underserved populations. Social determinants of health (SDOH) might have an impact on morbidity, favoring higher degrees of disability and even increasing mortality. The present study provides an overview of the global impact of MS now reported to affect most ethnicities in the world. Some epidemiological peculiarities and the existing equity barriers challenge access to modern diagnosis and treatment, particularly the utilization of advanced therapies.

## 2. MS Global Epidemiology in Relation with Societal Barriers to Care

The most comprehensive global compilation of MS information is assembled in the Atlas of MS [8] with data from the MSIF, the World Bank, the World Health Organization (WHO), and other global networks addressing different aspects of the disease. The WHO member states are grouped into six regions (alphabetically)**:** African, the Americas, Southeast Asia, Europe, the Eastern Mediterranean and the Western Pacific. Although MS has been reported in all these regions (Figure 1), prevalence rates vary from high to very low frequencies. Current data indicates that MS may affect the majority of the 7000 ethnicities populating the world (as the current paper shows), although there are reports of notable exceptions of virtual absence of disease in isolated non-genetically mixed South African Bantu, Inuit people in the Arctic region, American Indigenous groups, Samis in northern Scandinavia, Yakuts and other Siberian tribes and Australian Aboriginals.

African Region. North and South African countries exhibit prevalences between 5.0 and 20/100,000 inhabitants. Prevalence in Benghazi, Libya, it is reported as 5.9/100,000 inhabitants [9]. Egypt has the largest number of people with MS in the region (about 60,000) with a prevalence estimated at about 25/100,000, also characterized by younger age of onset and prompt development of greater disability [10]. MS data is still not available from 25 African nations including 10 newly formed sub-Saharan countries recognized by the UN. The vast majority of Africa’s inhabitants identify as black, displaying large ethnic, linguistic and cultural diversity. The North African countries are mostly of Arab-Berber ethnicity with linguistic and cultural ties with the Middle East. The literature addressing MS issues in Africa is limited but emphasizes the economic burden and societal impact exerted by the disease. Large portions of the African population do not have access to adequate diagnostic procedures for MS, ergo, appropriate management. In 2017, the African continent had the lowest density of MRI scanners at 0.7 per million people. This is compounded by the fact the ratio of trained neurologists is 0.03 per 100,000, in comparison to Europe’s ratio of 9.2 per 100,000. Public information and low awareness on the disease is lacking facilitating a cultural stigma and contributing to a substantially delayed diagnosis [11].

The Americas. The largest countries in North America, Canada and the United States (U.S.), are also among the areas with the highest MS frequencies in the world. The larger ethnic and racial groups in both countries are whites of European ancestry, 82.2% and 61.6%, respectively. Minorities in Canada are represented by 5% black people of African origin and 4.3% defined as First Nation Indigenous peoples. The national MS prevalence in Canada is reported at 290/100,000 inhabitants [12]. Specific prevalence of MS in Indigenous populations in Canada is estimated at 18.5/100,000 inhabitants, which although lower than the general Canadian population, is higher than other Indigenous populations outside Canada [13]. A caveat of this occurrence is that many of these groups have had long standing widespread genetic contact with Europeans, theoretically increasing their MS frequencies. This phenomenon is noted significantly among the Metis population: a distinct Indigenous people originated from mixed First Nations populations and European heritage. On the other hand, MS cases in the isolated Inuit people living in the Arctic and Sub-Arctic regions in North America and Greenland have not been documented despite rare, anecdotal non-published reports (i.e., blogs). The ratio of neurologists in Canada is estimated in 2.2/100,000 people, while in the U.S. is ~12.0/100,000 with some variations depending upon the population studied, i.e., regional or zonal, Medicare beneficiaries or people covered by private insurance. The U.S. reports the largest number of people living with MS: nearly 1 million, exhibiting a variable presence of the disease in different regions of the country, reaching a prevalence of over 300/100,000 in the northeastern and northwestern states, while demonstrating lower ratios in the southern and western areas: ≥270/100,000 [14], where the larger concentration of the minority groups, “Hispanic”/Latino (19.1% of the population) and black African Americans (12.4% of the population), reside. A patient data base (Kaiser Permanente Southern California) showed that the increased incidence of MS in African Americans was seen primarily in women [15]. Other reports show the age of onset of African Americans is 2.5 years later than in white “Caucasian” Americans of European ancestry [16]. Age of onset in Latino Americans living in the U.S. is reported to be younger than white “Caucasian” individuals and between 5 and 9 years younger than other ethnicities at the time of diagnosis [17]. Epidemiologic studies continue to be encouraged in the U.S. Despite the strong economic standing of these North American countries—the U.S. is number one in Gross Domestic Product (GDP) and Canada occupies the 10th position [18] in the world—gross societal inequities remain. The U.S. government reports 8.2% of the U.S. population not having health insurance [19], and while primary healthcare coverage in Canada is nearly universal (99.7%), 25% of the population do not have private secondary insurance coverage to assist in obtaining prescribed medications and costly tests like MR imaging. The Americas Committee for Treatment and Research in MS (ACTRIMS) was founded in 1995, promoting investigations and new therapies, and any development improving the knowledge of MS through the work of researchers and clinical neurologists in Canada and the U.S. has shown that “Hispanic”/Latinos and black African Americans in the U.S. have greater healthcare disparities and inequities than white populations [20]. This situation appears to affect large segments of these groups associated with pervasive low education and low socioeconomic levels. While studies on immigrants are scarce, there is indication that Mexicans with MS migrating to the U.S. (Los Angeles, California), or diagnosed after residing in the country, experience a late onset of disease in comparison to ethnic Mexicans born in California (34.2 years vs. 28.5; *p* < 0.001); the migrant group had also early higher disability (EDSS ≥ 6.0) in comparison to the California-born Mexicans (28% vs. 18%; *p* < 0.001) [21]. Whether this is a real biological phenomenon or due to an actual sociopolitical situation (i.e., language barriers, inadequate education and fear of deportation) remains to be established with further observations and studies. The term “Hispanic” has been criticized even though it is commonly used and officially utilized by the U.S. Census since 1970 to define populations of Latin American origin (including the Spanish-speaking Caribbean nations) “or other Spanish culture or origin regardless of race”. The term excludes non-Spanish speaking people, including Portuguese and Brazilian, and several groups from the Caribbean. ‘Latino’ was added to the descriptive term in 2000 [22]. The U.S. is the only country in the world utilizing the term “Hispanic”. Studies utilizing Genome wide association (GWAS) technology show native ancestry is associated with optic neuritis and age of onset in these populations [23]. Latin America (LATAM) is formed by 20 countries south of the U.S., including Mexico; Central America; the three Spanish-speaking Caribbean Islands, Cuba, Puerto Rico and the Dominican Republic; and the South American countries. The primary languages are Spanish and Portuguese, and the region is characterized by great racial, ethnic and cultural diversity. The predominant ethnicity are Mestizos, the result of five centuries of genetic admixture between Native Americans and Europeans. Biracial ethnicity from whites of European origin and black populations from African ancestries is also common in large areas of LATAM, resulting as well from centuries of genetic blend in the Americas.

The Atlas of MS shows the majority of the LATAM countries have low-to-mild frequencies, 0–25/100,000 inhabitants, except Argentina demonstrating moderate prevalence: 25–50/100,000 population. There are no data from Bolivia and the Guianas. Some studies from the Central American and the Spanish-speaking Caribbean region have recognized the emergence of MS as a public health problem, considering the realistic challenges to early diagnosis and treatment existing throughout the region. A compounding factor to this concern is the fact the neurologists rate in this area, involving eight countries, is reported to be very low with an average of 0.8/100,000 inhabitants [24]. World Bank data utilized by the Atlas of MS ranks LATAM countries within the low- and medium-income range. This reflects in substantial socioeconomic challenges affecting national programs and healthcare in general. These chronic and persistent challenges affect healthcare priorities and potential assignments of resources to assist in a costly emerging neurological disease in the region such MS. Care of people living with MS in LATAM basically depends on public healthcare and social security governmental programs. Private care is limited [25].

A common observation in LATAM is the lack of documented MS cases among genetically non-mixed Indigenous groups (Amerindians; Native Americans). These populations typically live in isolated communities without adequate access to healthcare and are often affected by limited education and information and subjected to racism. Indigenous populations in the American continent, including the Caribbean, account for large segments of the populations of each country, albeit demonstrating variable proportions probably related to internal national societal development, urbanization, miscegenation and other factors. There are at least 500 different Indigenous groups in the Latin American region [26]; the largest proportion is reported in Guatemala with 43.56% of a total national population of 18.4–18.7 million, although Mexico claims the largest population of Indigenous people: 23.2 million, 19.4% of a national total population of 130.9 million [27]. Brazil, the only Portuguese-speaking and the largest country in LATAM, discloses the smallest Indigenous proportion, 0.88%, out of a population of 212.8–213.4 million, mostly confined to the Amazon regions. Argentina, the only country in LATAM with a majority white population of European descent (Spanish and Italian), also demonstrates a small proportion of Amerindians (2.5%), although their numbers, languages and culture appear to be rapidly dissipating.

One of the early MS epidemiological studies in LATAM carried out in central Mexico in 1966 [28] did not identify the disease among the major Amerindian peoples living in that area. This finding reinforced the notion that MS was either extremely rare or not existing among LATAM Native Americans and stimulated interest to conduct additional epidemiologic studies in the region. Another study did not detect MS in non-mixed Amerindians in Chihuahua, a border state with the U.S. in northern Mexico [29], as well as a comprehensive multigenerational study performed on Lacandon people residing in the sierras of the southern state of Chiapas, bordering with Guatemala, Central America [30]. Epidemiologic studies in Panama [31] also did not reveal identifiable MS cases among the local Amerindian ethnic groups. This generalized observation has been reported in studies from Brazil, Colombia, Chile and Ecuador, and a consensus on this respect was emitted by the Latin American Committee for Treatment and Research in MS (LACTRIMS) in 2002 [32].

Eastern Mediterranean Region. The Middle East countries are traditionally included in this region constituted by 25 countries with 9 major languages and many dialects from the Levantine Arabic group spoken across the region. The MS prevalence fluctuates from very low to moderate-to-high with significant disparities in care related to several factors including income levels, active wars and a large number of refugees, particularly in Jordan and Lebanon, resulting from these conflicts. Displacement of people adds to poverty indexes and substantial limitations in healthcare. Early diagnosis of MS and utilization of DMT in these populations are severely affected. Despite these hazards, the prevalence in Lebanon is reported as 62.9/100,000, and in Palestine (West Bank) this number is 50/100,000 [33]. The Middle East and North Africa Committee for Treatment and Research in MS (MENACTRIMS) was founded in 2012 aiming to become a vehicle to promote epidemiologic studies, education and development of specialized care centers. Egypt is an active member of MENACTRIMS but listed in the African region by the WHO (see above) in view of its geographic location. Characteristically the Eastern Mediterranean countries have a high solar exposure theoretically resulting in adequate or high serum vitamin D levels among the populations. Low or deficient vitamin D levels are considered a risk factor for MS [34]. A common observation in the region is the notorious increase in MS incidence among women. The majority of Arab countries and non-Arab Iran are predominantly Muslim, hence wearing the hijab is considered compulsory for women after puberty. Low vitamin D levels in young females in Palestine have been adjudicated to strict dress codes: 7% of women (98% wearing the hijab), while the majority of males had optimal levels [35]. Iran claims the highest prevalence in this region (148/100,000), with variable frequencies throughout its territory (higher rates in Tehran and Isfahan), but increasing consistently throughout the country over the last decade. High MS prevalence in Iran appears to be closely related to a *sui generis* genetically heterogenous majority population (61% Persian; 16% Azerbaijani; 10% Kurdish). This augmented presence of MS has been particularly notorious among Iranian women, also reportedly associated with wearing the hijab along with loose-fitting clothes covering the arms and legs, inadvertently but effectively blocking solar exposure, resulting in insufficient vitamin D utilization. This dressing code became a legal requirement after the Islamic Revolution in 1979 (*Enquelãb-e Eslãmî*). The MS incidence in Iran increased significantly between 1989 (0.68) and 2008 (2.93), with female prevalence (Tehran’s data) increasing dramatically from 134/100,000 in 2014 to 252.65/100,000 in 2020. Concomitantly, during the same period of time, the male population prevalence of MS in Iran increased from 42.45/100,000 to 83.15/100,000, resulting in a female-to-male ratio of 3.016:1 [36]. Kuwait is another area reporting high prevalence, 105/100,000, with Palestinian emigres affected three times more than local Kuwaitis. Qatar reports a prevalence of 64.57/100,000 largely contributed to by the immigrant working force. The Kingdom of Saudi Arabia shows a medium-to-high prevalence 40.4/100,000 [37,38]. These countries are wealthy and provide sources of work for immigrants not just from neighboring countries, and more recently, from Palestine, Syria and Yemen, but also jobs for many Asian expatriates and even westerners occupying professional and technical positions. Some peculiarities described in Saudi studies indicate that associated neuropsychiatric disorders, namely severe depression, were more prominent in MS patients residing in the northern areas of the Kingdom, and that 13% were foreigners [39]. Two African countries located on or about the Horn of Africa, Djibouti and Somalia, are included by the WHO in this region. Djibouti shows a very low MS prevalence, 4.56/100,000; however, the true prevalence may be higher due to limited access to healthcare and diagnostic resources considering the country has a very low-income status. Somalia has one of the lowest reported MS prevalence rates in the world: 2.66/100,000 people [40]. There is no data from several countries from the Eastern Mediterranean region including two with large geographic extensions such Algeria and Syria, the latter affected by longstanding war conflicts, millions of refugees and internal displacement and widespread poverty. Cigarette smoking and *hookah* (waterpipe tobacco) vaping are widespread habits in this region, particularly among men, while obesity is more common among Arab women. All these are known factors to accelerate onset and worsening of the clinical course of MS. Significant disparities in MS care exist among all countries of this region, with a marked unequal access to modern diagnostic tools and therapy to fulfill current international criteria. These factors primordially affect countries overwhelmed by conflict or by low economic capabilities which are quite prevalent in the region. Due to considerable regional political discord, Israel stands by itself in this geographic area with a unique and complex demographic profile. The MS prevalence is reported as intermediate with a frequency of 95/100,000 inhabitants, affecting a rather distinct social composition (2024/2025) of 73.2% white Jews with origins from Europe and America; 21.1% Christian and Muslim Arabs and Bedouins; and the rest formed by the Druze and other groups. MS is more common among Ashkenazi Jews of European descent compared to Sephardic, Mizrahim and other Jewish groups, albeit these non-Ashkenazi people display a younger age at onset and more severe disability in general [41]. While universal funding for healthcare in Israel is obligatory, challenges subsidizing availability to specialized MS care notably exist in the country. This appears to be an obstacle affecting equitable access for low-income Israelis and ethnic minorities. Diplomatic difficulties have impeded Israel to be a member of MENACTRIMS, hence, the European committee (ECTRIMS), founded in 1985, has introduced this country within the organization and as member of the council. Pivotal studies from Israeli scientific works have resulted historically in the development of DMT such as Glatiramer Acetate (Copaxone^®^) [42]. Turkey, another country geographically located close to the Middle East and North Africa, has also been an active council member of ECTRIMS. Studies demonstrate intermediate-to-very high national prevalence of 105.2 per 100,000 in Afyonkarahisar and 288 per 100,000 in Silvas. The female-to-male ratio is 2.1:1, and there is generally a late onset of the disease, with an average age at diagnosis of 35 years [43]. Turkey’s healthcare system provides universal access through a general health insurance program. Even though the government has made a significant investment in its healthcare infrastructure, people with MS living in remote or rural areas have to travel substantial distances to receive attention in tertiary care institutions situated in larger cities. Their specialized neurologists are recognized with international experience, most trained in Europe, while their ratio to the population is within the range recommended by the WHO: 4 neurologists per 100,000 people. While access to DMT has been facilitated by the Turkey’s modern health system, reimbursement processes are complex and bureaucratic, despite the fact that MS therapies are promoted as cost-effective options for foreign patients seeking care in the country (“medical tourism”) [44].

Europe. The WHO European Region serves 53 countries geographically forming the western peninsula of the supercontinent Eurasia, including the largest country in the world: Russia. In 2020, 1,188,000 people lived with MS in Europe. The continent is commonly divided into subregions in the context of cultures and compass points: Western, Nordic, Central and Southern Europe. Some of these countries, in addition to having the highest reported MS frequencies according to the Atlas of MS from the MSIF (101–200 and even >200/100,000 inhabitants), are ranked by the World Bank as upper-middle or high income. In this group are the UK, Germany, Scandinavian countries, France and Italy. The highest prevalence rates are disclosed in the Scottish Northern Islands, which include the Orkney (402/100,000) and Shetland Islands (295/100,000) [45], and mainland (Highlands) Scotland (376/100.000) [46]. Moldova is considered by the MSIF as a country with moderate prevalence (37/100,000) and is also the nation with the lowest frequency reported in Europe. After the dissolution of the USSR in 1991, several countries formed in Central and Eastern Europe. This geopolitical phenomenon was further contributed by the breakup of Yugoslavia into Bosnia and Herzegovina, Croatia, Macedonia, Slovenia, Serbia and Montenegro. Slovenia and the Czech Republic have particularly improved their national economic status during this process, resulting in better opportunities for care for people with MS. The disease has an important epidemiologic presence in this part of the world and appears to be associated with increasing economic affluence. Some countries in Eastern Europe and the Balkans are still experiencing a complex, on-going political and economic development processes, but report high MS prevalences (>100/100,000); a few nations demonstrate an intermediate range (51–100/100,000 population), but no country from this area reports lower-than-moderate frequencies, meaning less than 26 MS cases per 100,000 inhabitants according to the rates utilized by the MSIF [47]. The Baltic States encompassing Estonia, Latvia and Lithuania in Eastern Europe, also resulted from the USSR break up, continue to improve their economic status and characterization studies on MS. Lithuania, considered to have the higher income, also reports a high MS prevalence estimated on 1 case per each 1000 people, female-to-male ration of 2:1 and annual incidence of 13/100,000 [48]. The Russian Federation reports variable MS intermediate prevalence from 10 to 80 cases per 100,000, depending on region and ethnicity. Significant gaps in access to appropriate diagnostic tools and disease management exist between people with MS residing in urban and rural areas. Western Siberian populations are of white European (“Caucasian”) genetic composition and demonstrate an increase in prevalence over three decades: 24 to 54/100,000 [49]. On the other hand, MS frequencies among Yakuts and other Siberian tribes residing in the easternmost remote areas of Siberia remain unreported [50], particularly among genetically non-mixed individuals. These communities commonly do not have as much access to optimal MS management as the western-located communities. The Russian Federation has a mandatory insurance health system and an active MS civil association previously called “All-Russian Public Organization of MS Patients” now named the Russian MS Society (RuMSS) [51]. The disease has been recognized as a national social and economic priority. Access to commonly utilized MS DMT sanctioned in the west by the US Food and Drug Administration (FDA) and by the European Medicines Agency (EMA) are not widely utilized in Russia. There are some locally produced ß-interferons 1a and 1b; however, modern data show these are low-efficacy treatments for relapsing MS, while numerous moderate- and high-efficacy, more advanced therapies, are being utilized in western countries. A peptide-bound liposomes drug, Xemnus^®^, a drug developed by Russian scientists, is produced with a relatively lower cost than other DMT and is provided free of charge to MS patients [52]. There have been no confirmatory clinical trials performed in the west on this therapeutic proposal. Neurological clinics in Russia offer inexpensive Stem Cell Therapies (SCT) for MS utilizing *sui generis* protocols subjected to insufficient regulations and becoming popular locations for international “medical tourism” [53]. In the west, safety and efficacy data for SCT are not yet robust enough to justify its standard use as option in the current MS therapeutic armamentarium, sanctioned by the FDA and EMA. The Russian Federation actively participates in the MSIF and ECTRIMS activities. The official language of these organizations in English. The European Union (EU) developed in 1993 encompasses most of the countries of the region as a political and trade alliance and economic convergence, communicating in 12 languages. Coordinated efforts from the EU include its European Commission on Health & Consumer Protection Directorate General and support international MS study networks such as the European MS Platform (EMSP) [54]. ECTRIMS has consolidated itself as the largest multinational organization in the world studying research and therapies for MS. The ratio of neurologists in Europe is 9.2 per 100,000. Despite the many areas of progress benefiting this region, home of the densest MS populations, important limitations and marked disparities in care remain in Europe. These issues are identified by the EMSP as unequal access to treatment emphasized by the fact that less than 60% of people with MS received DMT in 2020. Fragmented support is noted especially for people with progressive disease, and there is across the board lower availability and reimbursement for treatments and management of symptoms. Strengths and weaknesses in MS Care in Europe assessed by the “MS Barometer” of the EMSP, involving utilization of therapies, research, employment and social support, rehabilitation, establishment of MS registries and other elements, appear to be closely associated with national GDP in most cases. The two groups identified with very low MS frequencies include the Romani, the largest transnational minority ethnic group, and the Finno-Ugric (Sami) in Finland, despite this latter people residing in an area of high prevalence. Both groups are recognized as barely or non-genetically intermixed. A recent great sociopolitical challenge in Europe is the arrival of large displaced populations, refugees and migrants often from low MS prevalence countries coming to these new high-risk areas. The immediate and long-term epidemiologic MS consequences have not been properly evaluated.

Southeast Asian Region. This region is formed by 11 countries exhibiting a great variety of cultures, languages and ethnicities. MSIF rates the prevalence in the area as very low (i.e., Thailand 0.77/100,000) to generally low (~8 to 9 per 100,000), although substantial inter-regional differences exist. India, the largest country in the Southeast region and the most populous country in the world (1.419 billion), reports a national prevalence of 7 to 10/100,000 inhabitants, i.e., 8/100,00 in Mangalore, South India [55], while the small national Zoroastrian Parsi ethnic community reportedly has a higher prevalence of ~26/100,000 [56], the majority residing in Mumbai. The World Bank views India as a major, rapidly growing middle-income economy, ranking in the fifth place among the largest economies in the world. There is a rather active MS Society of India (MSSI) [57] with numerous chapters distributed throughout the Indian subcontinent providing support for the patients and caregivers and engaged in promotion of epidemiologic studies [58]. Despite the optimistic economic forecast for India, there are important limitations in access to care, particularly in rural areas. The general feeling is that greater awareness, more infrastructure for rehabilitation and specialized MS clinics, government support and affordability of DMT are fundamental aims for the general management of the disease in the country [59]. The Republic of Korea is included by the WHO in this region. This country has developed important advances in healthcare in the course of modernization of its economic structure and organization, including eligibility for government’s financial support for medical care and use of some DMT imported specifically for people diagnosed with MS, despite the low prevalence of the disease: 3.5/100,000. Many of these countries, experiencing a dynamic industrial growth, formed the Association of Southeast Asian Nations (ASEAN), a regional organization for economic, political and security cooperation. This environment of stability favored the development of the Pan Asian Committee for Treatment and Research in MS (PACTRIMS) in 2007. Common concerns and observations regarding MS in the Southeast Asia region refer to the consideration of this entity as a “rare disease” resulting in inadequate focus assigning resources for management and posing important differential diagnostic challenges.

Western Pacific Region. One quarter of the population of the world lives in this region. The WHO lists 38 nations in this region spanning from Central Asia to the Pacific Ocean, including nations as far north as Mongolia to as far south as New Zealand, as well as eleven Pacific Island countries. The ethnic and racial variety of the populations living in these areas are the most diverse genetically in the world, while the population residing in the region is about 2.2 billion. The Atlas of MS of the MSIF assigns a general low prevalence (5/100,000) to the Western Pacific areas; however, the two countries with the high ratio of white people of European origin in this region, Australia and New Zealand, both report prevalences of about 100 per 100,000.

The Indian diaspora comprises the largest migratory phenomenon in the world, including resettling in neighboring countries in the region. A pivotal MS epidemiologic study in Malaysia showed the Malays (52.9%) were the predominant ethnic group followed by the Chinese and Indians, all of whom were from West Malaysia, while a small proportion (1.9%) was comprised by Indigenous subjects from Sabah and Sarawak from East Malaysia. The MS female to male ratio is reported in 5:1. In these series, the investigators carefully ruled out Neuromyelitis Optica Spectrum Disorder (NMOSD) utilizing strict clinical criteria and positive serological anti-aquaporin antibody, particularly in people exhibiting longitudinally extensive spinal cord lesions on MR images. NMOSD is a rare CNS inflammatory disorder, different from MS in pathophysiology and therapeutic management and common in Asians. In this study, a subgroup analysis revealed more Chinese people having NMOSD rather than MS [60].

Comparative studies across three main genetically diverse Asian ethnic groups in Singapore show higher MS frequencies among its so-called ‘South Asian population’, followed by Malays and Chinese [61]. Considering the great ethnic admixture in Singapore, ethnic identification has been rather challenging, even confusing. The national Ministry of Home Affairs (MHA), in order to maintain “racial harmony”, instituted a race model called ‘CMIO’ for Chinese-Malay-Indian-Others to define the local racial framework. Nevertheless, genetically, this population complex has been shown to be 75.5% Chinese, indicating the high presence of this genetic group in Singapore [62]. China (People’s Republic of China) is one of the most populous countries in the world (1.407 billion); it covers a huge geographic extension and it is the second wealthiest nation. China reports a very low MS prevalence, 2.32 per 100,000, with a slow increment from 1990 to 2010 and increasing tendency during the last decade [63]. MS literature from China is scarce. Data from the Shanghai Statistical Yearbook (2004) estimated the raw MS prevalence in the province at 1.39/100,000 population [64]. Information from the south China areas Hong Kong and Taiwan showed a prevalence (2016) of 0.77/100,000 population [65]; however, more recent data from Hong Kong discloses a pronounced increased in frequency: 6.9/100,000 [66]. In general, the female-to-male ratio is reported 3.0–3.2:1. The China’s State Organizational Structure (CECC) [67] oversees the China’s National Medical Products Administration (NMPA). This office has released seven DMT for people diagnosed with MS: Interferons 1a and 1b, Teriflunomide and Dimethyl Fumarate, all considered of low-to-moderate efficacy; two medicines of the selective sphingosine 1 phosphate-1 receptor modulator class: Siponimod and Fingolimod, both of moderate efficacy; and ofatumumab, a B-lymphocyte depleting agent of high efficacy. In comparison, the current MS therapeutic armamentarium of occidental countries, including western Europe, the U.S. and Canada, counts with more than 20 approved medications by the FDA and EMA. The DMT available in China are provided to people diagnosed with MS through coverage from the multifaceted Chinese healthcare system aimed to achieve universal protection, but still a great part of the cost is the responsibility of the individual. The concept of third party or private insurance does not exist. Lack of more actual and precise MS epidemiology data in China appears to be an important barrier to the development of more efficient healthcare policy. The MS prevalence in Mongolia, a geographically vast country of 3.5 million in the center of the Asian continent, remains unknown since there are no studies on epidemiology and clinical characterization of the disease, and no DMT is available. One report explored predictors of disability and depression in 27 Mongolian patients; the majority (88.9%) had moderate-to-severity disability (median EDSS score, 5.5) [68].

There have been periodic epidemiologic studies in Japan, particularly from the north of the archipelago, showing over the last two decades a slow increase in incidence rates of 0.99/100,000, prevalence of 22.4/100,000 and female-to-male ratio of 4.0:1 [69]. Long-standing barriers faced by Japanese MS patients include limited neurological specialized care along with high costs of DMT. Limited public education on MS favors endemic social stigma from the disease and lack of loss of employment protection in Japan [70].

The Commonwealth of Australia, constituted by the large continental landmass and the Island of Tasmania, reports an increasing MS prevalence of more than double within a documented period of 11 years: from 99.5/100,000 in 2010 to 131.1/100,000 in 2021 [71]. These people concentrate in the southeast areas of the country where the larger cities and more populated communities are located. The majority of the Australians are whites of European ancestry, mostly from the United Kingdom (UK) and countries from the western European continent. There is increasing immigration from India, China and the Philippines, but thus far it has not made an impact on the Australian national MS frequencies. Aboriginals and Torres Strait Islanders form about 3.8% of the general population, but the prevalence of MS in these groups is negligible to the point that it has not been explicitly quantified. Risk factors contributing to longitudinal increases in MS prevalence in Australia have been recognized, including vitamin D deficiency, smoking rates in females, childhood and adolescent obesity and late first pregnancy [72]. An issue of concern in Australia is the cost affecting a person living with MS in the country, which is substantially higher than other comparable complex chronic diseases such as Parkinson’s Disease, Type 2 diabetes and long-term cancer [73]. New Zealand (or Aotearoa in Māori) is formed by two land masses (North and South Islands) and a multitude of smaller islands. The northern lands are the most populated, with 85% of New Zealanders living in urban areas in 10 cities. The majority of the population, 62.1%, are of European ancestry, mostly from the UK; 17.8% identify as Māori; and the rest of the demographic is constituted by Chinese, Indians, Samoans, Filipinos and Tongans. Typically, Māori people are ethnically mixed in New Zealand. Since the colonization era by the British in the nineteenth century, intermarriage between Māori and people of other ethnicities, particularly European, has taken place. It is common for the Mãori to identify dually as European (46.1%). Recent epidemiologic studies in New Zealand have shown an increase from a prevalence of 72.4/100,000 in 2006 to 98.6 per 100,000 in 2022. This increase has been particularly noted among European New Zealanders at the rate of 132.3 per 100,000, while Māori rates rose from 15.0 to 33.1 per 100,000 during the 12-year period comprehended in the report [74]. Despite the increasing MS rates among the Māori, the frequencies remain about 4-fold less than the white population with MS. Although the gradual MS rate augmentation among the Māori could be adjudicated to their European genetic admixture, an earlier study performed in Wellington, New Zealand, had shown that HLA A3, B7 and DR2, found in European local subjects with MS, were less common in Māori individuals with MS [75]. Nevertheless, genetic studies among MS groups in the country have not been duplicated or pursued. The last study showed a female-to-male ratio of 3:1. The biggest barriers to diagnosis in New Zealand are the lack of MS awareness in the community and critical shortages of specialist services and MRI equipment, despite the government efforts to alleviate healthcare limitations in the country (9.2% of the GDP is spent in healthcare). Māori are generally over-represented in measurements of socioeconomic deprivation in the country. The lower risk of MS among those with low socio-economic status (“deprived”) possibly reflects to a certain degree difficulties in accessing prompt diagnosis and therapy [76]. DMT are available, but guidelines can be more restrictive compared to other developed countries.

## 3. Discussion

This opinion offers an overview of the status of MS in the world, addressing epidemiological aspects and barriers in equity of care accompanying its presence in the different regions of the globe. Current knowledge indicates the disease affects most people in the world regardless of race and ethnicity, albeit marked epidemiological variability exists. This overview concentrates on prevalence data in the view that information on incidence rates is inconsistently reported. The disease is now increasingly identified among populations previously considered to be uncommonly affected or to have a very low risk of developing MS. The reason for this development is not completely understood, although several risk factors have been invoked including genetic dissemination. This phenomenon appears to have favored the appearance and increment of MS in people product of the intermixing of high-risk groups with low-risk populations, resulting from historical geo-political events favoring racial amalgamation. The common MS European inherited genetic marker HLA-DRB1*15:01 is particularly rooted in Scandinavian and northern Europeans, carried by 50% to 60% of their populations with MS, and it is present almost in one half of other white populations with MS, including U.S. Americans [77].

The historical introduction of the European HLA-DRB1*15:01 to traditionally less susceptible groups has been particularly noted among Latin American Mestizos and biracial people of African ancestry [78]. Some of the risk factors identified to trigger or worsen disease, such as smoking and childhood and adolescent obesity, appear to be associated in some countries with educational levels. In the Middle East countries, smoking is a widespread social habit among men, while obesity is more common among women. Although vitamin D deficiency is a MS risk factor commonly associated with geographic latitude, (Canada and northern Europe located at 45–60° N; Australia and New Zealand at 30–45° S), its increasing prevalence in covered Muslim women in the Eastern Mediterranean Region (25–39° N) is of particular interest. Globally, the female-to-male ratio is consistently 2.0–3.0:1, with increasing rates in women noted among Asian and Middle East populations.

Social determinants of health (SDOH) play a significant role as the non-medical factors globally influencing MS. Income and employment status are specially affected, particularly in people with the progressive forms of MS or due to cognitive dysfunction produced by the disease. Most healthcare systems, even from developed high-income countries, do not have well-designed or operational programs for the care and protection of disabled people with MS. The European MS Platform reports that 24 countries do not have neurological or chronic disease policy which includes MS, and that 52% of people with MS in Europe do not have access to physical rehabilitation services. These drawbacks exist to a certain degree in most countries included in the present study, particularly the ones ranked as low-income nations. A potential model for social distress awareness is New Zealand’s Deprived Area Index incorporated since 1991 in the national census [79]. This “index of deprivation” provides indicators on income, employment, education qualifications, housing, etc. The data show that Māori people with MS in New Zealand are over-represented in the measurements of socioeconomic deprivation reflecting obstacles to access healthcare in general, early diagnosis and therapies, particularly access to the expensive new therapeutic molecules. The European MS Platform reports that 43% of people with MS in Europe did not receive DMT in the last five years.

Across the globe, people with MS rely on their government-sponsored health programs for DMT acquisition, either through taxation to support social security or obligatory health insurance. Inability to afford medicines is the most common reason for not utilizing MS treatments in close to 90% of low- and lower-middle-income countries according to data from the World Bank and the WHO [80]. It is evident that in countries with higher national income, more availability of medications exists, and, hypothetically, larger portions of the population with MS potentially may have access to expensive DMT.

The current therapeutic armamentarium is constituted by DMT with a great variability in efficacy. All these medicines have been licensed by the main international health agencies, the Food and Drug Administration (FDA) in the U.S. and the European Medicine Agency (EMA), after showing efficacy through evidence-based clinical trials. Both agencies serve as international examples for approval of new or complex therapeutic molecules. The first DMT (also called platform injectables), interferon ß 1-a and 1-b, were approved in 1993, followed by glatiramer acetate. Numerous oral agents were licensed starting in 2011 with the first oral DMT, fingolimod, followed by other medicines of the same class, sphingosine-1-phosphate receptor 1 modulators including siponimod, ponesimod and ozanimod, each displaying a similar mechanism of action and slight variations in the basic molecule. Other oral medications include teriflunomide, dimethyl and monomethyl fumarate, each having a different mechanism of action. While these oral agents have a moderate-to-high efficacy level, the oral DMT with the highest efficacy is a synthetic immunosuppressing agent: cladribine. The pharmacological medications with the highest degree of efficacy are the monoclonal antibodies (MAB), natalizumab (α4-integrin inhibitor), alemtuzumab (anti-CD52 antibody) and the group of ant-CD20 B-cell depleting antibodies: ocrelizumab, ofatumumab, and ublituximab. A generic anti-CD20 product, Rituximab, even though it is not officially licensed by the FDA and EMA, is widely used in the world as less expensive alternative. Natalizumab also has a follow-on or generic form approved by the FDA and EMA: tyruko. All these DMT are indicated for relapsing MS except ocrelizumab, also approved for primary progressive disease. The fact that a DMT is licensed and registered in the health system of a country does not mean the person living with MS in that area will necessarily have the means or support, personal or institutional, to receive the treatment.

The historical sequence of DMT approval by the FDA and EMA is shown in the diagram illustrated in Figure 2. The milestones are grouped into four epochs: historic era, the advent of MAB, the oral medicines era and the high-efficacy anti-CD20 MAB era.

There is a substantial lack of uniformity in availability of approved DMT in the world, mostly due to national economic factors, but also affected by lack of health policy for MS. Examples illustrating this situation are presented from Western Europe, U.S., Mexico, China, Russia and Panama (Table 1).

As an effort to achieve universal health coverage, a select committee from the MSIF provided the WHO with three medicines for the list of “Essential Medicines List” (EML): oral cladribine, glatiramer acetate subcutaneous injections and rituximab per IV infusion [81]. The EML is updated every two years. All these medications are available in generic form therefore the cost may potentially be reduced enough to be accessible to most places, even though many international experts have questioned this DMT selection. Some low-income countries in LATAM (i.e., Cuba, El Salvador and Nicaragua) have very limited accessibility to DMT. In many regions of the world, limitations in accessing therapy are compounded by reduced availability of neurologists and diagnostic tools. While these deficits may be generally associated with national GDP level, other local, individual, logistic and economic factors play a role. Inadequate census and insufficient epidemiological infrastructure favor under-reporting from many places.

## 4. Conclusions

Disparities in access to healthcare of people with MS are documented globally, fostered by increasing identification of the disease in all regions of the world affecting practically all ethnicities. The lack of epidemiological data has reflected substantial barriers to enable development of government policies addressing MS care in many countries of the world. The present opinion has the limitations of addressing aspects of inequity in MS care from a restricted number of countries with the view of limited available data. The era of considering MS as an exotic, uncommon disease is no longer a viable concept for most of the regions of the world.

## 5. Future Directions

Collective international progress improving public education and awareness on MS is required. An international path to alleviate lack of accessibility to the costly DMT medicines is necessary. This could only be accomplished with positive interaction among the pharmaceutical industry, the WHO, the World Bank, and other stakeholders such as local governments and national MS associations. The development of highly effective therapeutic products for MS, i.e., monoclonal antibodies, advanced anti-B-cell lymphocyte depletory agents, Bruton Tyrosine Kinase Inhibitors and other molecules should enable a realistic modification of the course of the disease. This should also reflect in a more positive prognosis for the MS patient.

## Figures and Tables

**Figure 1 neurolint-18-00002-f001:**
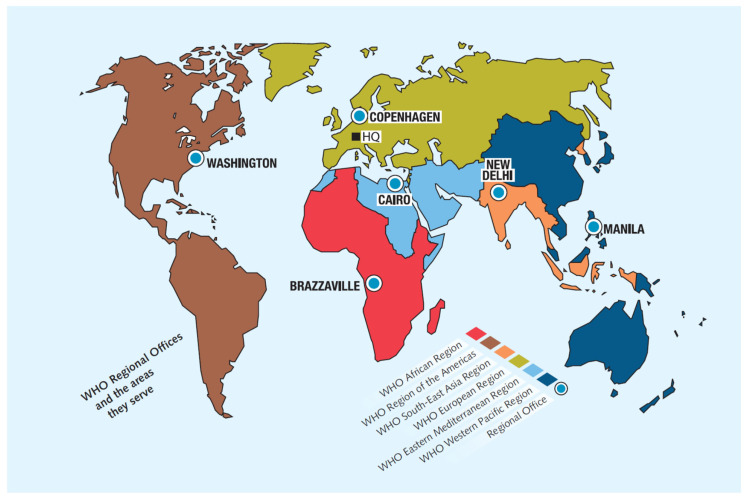
The World Health Organization regions. Regional offices (modified from Atlas of MS 2024). Source: Multiple Sclerosis International Federation [4].

**Figure 2 neurolint-18-00002-f002:**
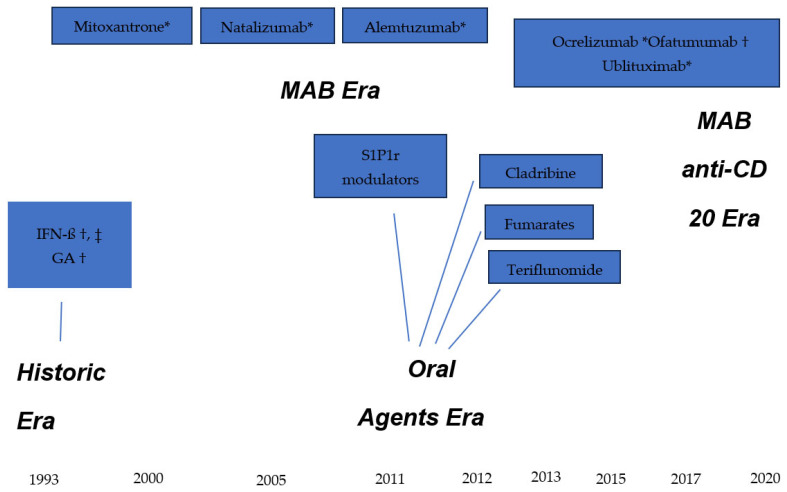
Therapeutic armamentarium for Multiple Sclerosis. * Intravenous infusion; † subcutaneous injection, ‡ intramuscular injection. IFN-ß = interferon beta; GA = glatiramer acetate; S1P1r = sphingosine-1-phosphate 1 receptor modulators.

**Table 1 neurolint-18-00002-t001:** DMT availability in selected regions/countries.

DMT	Administration Route	Western Europe *	United States *	Mexico *	China *	Russia *	Panama *
Interferonß 1-a and 1-b	IM SC	X	X	X	X		X
Glatiramer Acetate	SC	X	X	X			
DMF; MMF	Oral	X	X	X			
Teriflunomide	Oral	X	X	X	X		X
S1P1r modulators	Oral	X	X	X	X		X
Cladribine	Oral	X	X	X			X
Mitoxantrone	IV infusion	X	X	X			
Natalizumab	IV infusion	X	X	X			X
B-cell depleting MABs	IV infusion and SC	X	X	X	X		X

* Medication available through government and national social security systems or private insurance. IM = intramuscular, SC = subcutaneous, IV = intravenous. DMF = dimethyl fumarate; MMF = monometthyl fumarate; S1P1r = sphingosine 1 phosphate receptor (modulators: fingolimod, siponimod, ozanimod, ponesimod); MABs = monoclonal antibodies (B-cell depleting: ocrelizumab, rituximab, ofatumumab, ublituximab. These agents are administered intravenously, except ofatumumab which is injected subcutaneously).

## Data Availability

No new data were created.

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
