# Peer review of "Multiple Sclerosis: An Ethnically Diverse Disease with Worldwide Equity Challenges Accessing Care"

_2035-8377, 2025, doi:10.3390/neurolint18010002_

Round 1
Reviewer 1 Report
Comments and Suggestions for Authors
This is an opinion paper about the global status of multiple sclerosis epidemiology and healthcare systems, offering a unique perspective. However, individual sections appear to have room for improvement. For example, reference 64 cited for China's epidemiology is a review paper synthesizing data from various countries, yet it lacks citations to the original epidemiological data sources. Consequently, the reliability of the epidemiological data is questionable (it has zero citations to date). Caution is advised to avoid citing low-quality papers that could mislead readers.
Some references are outdated. For example, only two Japanese studies from 2003 and 2013 are cited. But there are newer papers.
Houzen, H., Kano, T., Kondo, K., Takahashi, T., & Niino, M. (2023). The prevalence and incidence of multiple sclerosis over the past 20 years in northern Japan. Multiple Sclerosis and Related Disorders, 73, 104696.
Watanabe, M., Isobe, N., Niino, M., Nakashima, I., Matsushita, T., Sakai, Y., ... & Research Committee of Neuroimmunological Diseases in Japan. (2024). Prevalence of, and disability due to, multiple sclerosis and neuromyelitis optica spectrum disorder in Japan by the fifth nationwide survey. Neurology, 103(10), e209992.
A re-evaluation of the cited literature and investigation of the latest literature is considered necessary.
minor points
Repetition of African region section
Author contribution: not complete?
Author Response
-This is an opinion paper about the global status of multiple sclerosis epidemiology and healthcare systems, offering a unique perspective. However, individual sections appear to have room for improvement. For example, reference 64 cited for China's epidemiology is a review paper synthesizing data from various countries, yet it lacks citations to the original epidemiological data sources. Consequently, the reliability of the epidemiological data is questionable (it has zero citations to date). Caution is advised to avoid citing low-quality papers that could mislead readers.
-Some references are outdated. For example, only two Japanese studies from 2003 and 2013 are cited. But there are newer papers.
Houzen, H., Kano, T., Kondo, K., Takahashi, T., & Niino, M. (2023). The prevalence and incidence of multiple sclerosis over the past 20 years in northern Japan. Multiple Sclerosis and Related Disorders, 73, 104696.
Watanabe, M., Isobe, N., Niino, M., Nakashima, I., Matsushita, T., Sakai, Y., ... & Research Committee of Neuroimmunological Diseases in Japan. (2024). Prevalence of, and disability due to, multiple sclerosis and neuromyelitis optica spectrum disorder in Japan by the fifth nationwide survey. Neurology, 103(10), e209992.
A re-evaluation of the cited literature and investigation of the latest literature is considered necessary.
REPLY: I am deeply grateful for the input and observations to improve the manuscript. The following actions have been carried out in response to your comments:
(1) Reference # 64 has been replaced with an updated specific citation of a paper on China's MS prevalence and published in Neurology in 2024. The data is discussed in lines 510 and 511.
(2) Thank you so much for providing references on more recent studies from Japan. I replaced reference # 70 with the more updated citation you kindly provided. The data is discussed in lines 536-538.
-Repetition of African region section
(3) The duplicated segment (most likely a technical glitch) has been deleted (lines 126-142).
Author contribution: not complete?
(4) The 'Author Contributions' (line 704) has been completed.
Reviewer 2 Report
Comments and Suggestions for Authors
The author has conducted a thorough literature review to present a clear and detailed overview of the global epidemiology of multiple sclerosis (MS), alongside the associated realities of diagnosis, treatment access, and healthcare disparities across different regions and ethnicities. The manuscript is well-structured, informative, and provides a valuable perspective on the worldwide challenges in MS care.
I have only one minor suggestion for potential enhancement:
The manuscript effectively lists the numerous disease-modifying therapies (DMTs) available and discusses the evolution of treatment options. To make this historical progression even more intuitive and impactful for readers, it might be beneficial to include a visual timeline (figure) illustrating the chronological approval (by the FDA and/or EMA) of key DMTs. This graphic could organize drugs by their approval date, categorizing them perhaps by administration route (e.g., injectable platforms, oral agents, monoclonal antibodies) or mechanism of action. Such a figure would provide an immediate, at-a-glance understanding of the rapid therapeutic advancements over the past three decades and help contextualize the current treatment landscape discussed in the text.
Author Response
The author has conducted a thorough literature review to present a clear and detailed overview of the global epidemiology of multiple sclerosis (MS), alongside the associated realities of diagnosis, treatment access, and healthcare disparities across different regions and ethnicities. The manuscript is well-structured, informative, and provides a valuable perspective on the worldwide challenges in MS care.
I have only one minor suggestion for potential enhancement:
The manuscript effectively lists the numerous disease-modifying therapies (DMTs) available and discusses the evolution of treatment options. To make this historical progression even more intuitive and impactful for readers, it might be beneficial to include a visual timeline (figure) illustrating the chronological approval (by the FDA and/or EMA) of key DMTs. This graphic could organize drugs by their approval date, categorizing them perhaps by administration route (e.g., injectable platforms, oral agents, monoclonal antibodies) or mechanism of action. Such a figure would provide an immediate, at-a-glance understanding of the rapid therapeutic advancements over the past three decades and help contextualize the current treatment landscape discussed in the text.
REPLY I do thank you very much for your comments and positive assessment of this work. I have followed your suggestion and included a Figure 2 with a diagram depicting the milestones of the historical advent of FDA and EMA approved medicines for the international MS therapeutic armamentarium.